# Cereal- and Fruit-Based Ethiopian Traditional Fermented Alcoholic Beverages

**DOI:** 10.3390/foods9121781

**Published:** 2020-12-01

**Authors:** Eskindir Getachew Fentie, Shimelis Admassu Emire, Hundessa Dessalegn Demsash, Debebe Worku Dadi, Jae-Ho Shin

**Affiliations:** 1College of Biological and Chemical Engineering, Addis Ababa Science and Technology University, Addis Ababa 16417, Ethiopia; eskench@gmail.com; 2Department of Applied Biosciences, Kyungpook National University, Daegu 41900, Korea; 3School of Chemical and Bio-Engineering, Addis Ababa Institute of Technology, Addis Ababa University, P.O. Box 385, King George VI Street, Addis Ababa 16417, Ethiopia; shimelisemire@yahoo.com (S.A.E.); hundessad@gmail.com (H.D.D.); 4Department of Food Engineering and Postharvest Technology, Institute of Technology, Ambo University, Ambo 2040, Ethiopia; debeworku2010@gmail.com

**Keywords:** traditional alcoholic beverage, Ethiopia, processing, physicochemical, fermentative microorganisms

## Abstract

Traditional fermented alcoholic beverages are drinks produced locally using indigenous knowledge, and consumed near the vicinity of production. In Ethiopia, preparation and consumption of cereal- and fruit-based traditional fermented alcoholic beverages is very common. *Tella*, *Borde*, *Shamita*, *Korefe*, *Cheka*, *Tej*, *Ogol*, *Booka*, and *Keribo* are among the popular alcoholic beverages in the country. These beverages have equal market share with commercially produced alcoholic beverages. Fermentation of Ethiopian alcoholic beverages is spontaneous, natural and uncontrolled. Consequently, achieving consistent quality in the final product is the major challenge. Yeasts and lactic acid bacteria are the predominate microorganisms encountered during the fermentation of these traditional alcoholic beverages. In this paper, we undertake a review in order to elucidate the physicochemical properties, indigenous processing methods, nutritional values, functional properties, fermenting microorganisms and fermentation microbial dynamics of Ethiopian traditional alcoholic beverages. Further research will be needed in order to move these traditional beverages into large-scale production.

## 1. Introduction

Worldwide production and consumption of fermented beverages has a long history, and is believed to have started around 6000 BC [1,2]. Production techniques and consumption of these traditional beverages are very localized [3]. Ethiopia, like other parts of the world, produces and consumes a significant volume of traditional alcoholic beverages (Table 1). About eight million hectoliters of Ethiopian traditionally fermented alcoholic beverages are produced yearly. Commercially and traditionally produced alcoholic beverages have an almost equal market share [4] and annual per capital pure alcohol consumption in the country is about 2 L [5].

Traditional alcoholic drinks are widely produced and consumed in Asia and Africa [6]. Rwanda’s ikigage [7], Nigeria’s oti-oka [8], Uganda’s kwete [9], Kenya’s Busaa [10], Korea’s makgeolli [11] and Mexico’s pulque [12] are among the most common traditional alcoholic beverage that are consumed and produced in each respective country.

In Ethiopia, *Tella* [1], *Borde* [13], *Shamita* [14], *Korefe* [15], *Keribo* [16], *Cheka* [17], *Tej* [18], *Ogol* [19] and *Booka* [20] are very popular indigenous fermented alcoholic beverages. The total alcohol content of these beverage is in the range of 1.53–21.7% (*v*/*v*) [18,20]. All of these Ethiopian alcoholic beverages are produced at a small scale and sold by local alcohol venders from their homes. These traditional alcoholic beverages are classified under the category of acid-alcohol fermentation systems [21].

Scholars define wine and beer based on various perspectives. For instance, Herman [22] defined wines as alcoholic beverages made from sound ripe grapes, whereas Pederson [23] defined alcoholic beverages based on the kind of substrates: beers are produced from cereals whereas wines are produced from fruits. In addition, Steinkraus [24] defined wine as an alcoholic beverage that uses sugar as the principal source of fermentable carbohydrate. According to Steinkraus [24], beverages made from honey, sugar cane and palm are classified under the category of wine. Hence, *Tej*, *Ogol* and *Booka* can alternatively be called wines, since honey is used as a major substrate for the fermentation process.

“Gesho” (*Rhamnus prinoides* L.), also known as “dog wood”, is the most common ingredient used to prepare Ethiopian alcoholic beverages, primarily as a flavoring and bittering agent. The substance β-sorigenin-8-O-β-D-glucoside (*“geshoidin”*) is the naphthalenic compound responsible for imparting bitterness [25,26]. In addition to this, it is also a source of fermentative microorganisms and plays a significant role during fermentation in regulating the microbial dynamic [27].

Due to the absence of standardized processes, back-slopping, and starter culture, Ethiopian beverages often have poor quality and failure to achieve their objective [28]. Moreover, preparation of these fermented alcoholic beverages is time-consuming and laborious [29]. As far as we know, this review is the first of its kind to address the research trends, significant research gaps and directions for future research outputs on Ethiopian traditional fermented alcoholic beverages. In particular, the raw materials, processing methods, physicochemical properties, nutritional values, functional properties, responsible fermenting microorganisms, fermentation microbial dynamics and storage stability of Ethiopian alcoholic beverages are the key points reviewed in the paper.

## 2. Cereal-Based Traditional Alcoholic Beverages

### 2.1. Tella

*Tella* is the most consumed traditional fermented alcoholic beverage in Ethiopia. It is the most popular beverage in the Oromia, Amhara and Tigray regions (Table 1). Barley, wheat, maize, millet, sorghum, “*teff*” (*E. tef*) and “*gesho*” leaves (*R. prinoides*) along with naturally-present microorganisms are the ingredients used to produce *Tella* [1]. Even though the volume of production and consumption is high, the fermentation process is still spontaneous, uncontrolled and unpredictable [41].

The *Tella* making process and its raw materials vary among ethnic groups and economic and traditional situations [37]. Although there are minor changes in the process in different localities, the basic steps are similar throughout the country. The making of “*Tejet*”, “*Tenses*” and “*Difdif*” are the fundamental steps in the *Tella* preparation process [1].

The *Tella* making process starts by soaking the barley in water for about 24 h at room temperature to produce a malt, locally called “*Bikil*”. After 24 h, the moistened grain is covered by using fresh banana leaves and kept in a dry place for an additional three days [39]. Then, the germinated barley grain is sun-dried and ground to produce malt flour. At the same time “gesho” (*R. prinoides*) leaves and stems are sun-dried and ground. Then, “*Bikil*” flour and “gesho” powder are mixed with an adequate amount of water in a clean and smoked traditional bioreactor known as “*Insera*”. This mixture is left to ferment for two days to form “*Tejet*” [31]. Subsequently, millet, sorghum and “teff” (*E. tef*) flours of equal proportion are mixed with water to form a dough. The dough is then baked to produced unleavened bread locally known as *“ye Tella kita”* [41], which is sliced into pieces and added to the earlier produced *“Tejet”*. The mixture is then sealed tightly to ferment anaerobically for 5 to 7 days to turned into *“Tenses”* [30].

While the *“Tenses”* is fermenting, maize grain is soaked in water for about 3 d, and then it is dried, roasted and ground to make a dark maize flour called *“Asharo”. “Asharo”* is the main ingredient that determines the color of *Tella* [31]. “*Asharo”* is then added to the previously produced *“Tenses”* and fermented anaerobically for a period of 10 to 20 days. After this period of fermentation, a thick mixture locally called *“Difdif”* is formed. Water is added to *“Difdif”* and left to ferment for an additional 5 to 6 h. Finally, solid residues are removed by filtration and served to consumers as *Tella*. In order to produce 25 to 28 L of pure *Tella*, 1 kg of *“gesho”* (*R. prinoides*) powder, 0.5 kg of “*Bikil”*, 5 kg of *“ye Tella kita”*, 10 kg of *“Asharo”* and 30 L of water are required [41].

Ingredients and utensils used to prepare *Tella* are the major source of microorganisms for the fermentation process [42]. As shown in Table 2, genera of *Saccharomyces*, *Lactobacillus* and *Acetobacter* are the most predominant fermenting microorganisms present in *Tella* [1,30,41]. The alcohol content and pH of *Tella* collected from different localities vary from 3.98–6.48% (*v*/*v*) and 1.52–4.99, respectively [43]. The alcohol content of *Tella* is greater than that of Rwanda’s ikigage [7] and is very much lower than Korean makgeolli [11]. The electric conductivity, salinity and total dissolved solids (TDS) of *Tella* are 2359 µs/cm, 1.2% and 1180 mg/L, respectively [44].

Since the production of *Tella* is performed at the household level, it seriously lacks aseptic processing conditions. Consequently, the shelf life is no longer than 5 to 7 days at room temperature. Beyond that, the flavor becomes too sour to drink. *Acetobacter* species are mostly responsible for this sourness because they convert ethanol to acetic acid in the presence of oxygen [45].

### 2.2. Borde

*Borde* is a cereal-based Ethiopian traditional fermented low alcoholic beverage that uses maize (*Z. mays*), wheat (*T. aestivum*), finger millet (*E. coracana*) and sorghum (*S. bicolor*) interchangeably or sometimes proportionally as the main ingredients [29]. It is commonly produced and consumed in the southern and western part of Ethiopia. The local communities consider *Borde* as a meal replacement. Particularly, low-income local groups of the population may consume up to 3 L of *Borde* per day [47]. The nutritional value is high due to the high number of live cells present in freshly produced *Borde* [32].

The *Borde* making process starts with germinating barley grain by following the same procedure described for the *Tella* malt preparation process. This malt, a source of amylase enzymes, is ground to become a malt flour [33]. In parallel, maize grits are mixed with a proportional volume of water and fermented for about 44 to 48 h (Figure 1). The fermented blend is divided into three portions. Similar to Uganda’s kwete [9], about 40% of the blend is roasted on a hot pan and a bread locally called *“Enkuro”* is produced. Then, the prepared *“Enkuro”* is mixed with malt flour and additional water and allowed to ferment for about 24 h in the same mixing tank [32]. The other 40% of the fermented maize grits are mixed with additional fresh maize flour and water. This mixture is shaped into a ball-like structure and cooked using steam to form *“Gafuma”* [29]. Subsequently, “*Gafuma”* is added to previously prepared *“Tinsis”* to become the thick brown mash called *“Difdif”* [13]. The remaining 20% of the fermented maize grits are mixed with additional flour and water and boiled to form thick porridge. Then, the prepared porridge, extra malt, and water are mixed into the earlier produced *“Difdif”*. Finally, the mixture is filtered and a small amount of water is added before serving to consumers as the final product *Borde* [33].

A good-quality *Borde* can be described as opaque, fizzy, of uniform turbidity, gray in color, with a thick consistency, a fairly smooth texture, and a flavor somewhere in the middle between sweet and sour [29]. The average pH values of *Borde* lie within the range of 3.6–4.1. The type of ingredients used and the processing conditions are the major causes for variation in the final product [32]. The conductivity, salinity and TDS values of *Borde* are 7139 µs/cm, 3.9%, and 3830 mg/L, respectively. As in Kenya’s busaa [10], yeast and lactic acid bacteria are the dominant microorganisms in *Borde*. Around 10^9^ CFU/mL counts have been recorded for both mesophilic bacteria and lactic acid bacteria [47]. In addition, a 10^5^–10^7^ CFU/mL yeast count has been reported for freshly prepared *Borde* (Table 3). Due to these high microorganism counts, *Borde* becomes unfit for consumption after 12 h of room temperature storage [29].

### 2.3. Shamita

*Shamita* is another traditional low alcoholic beverage that is produced and consumed in different parts of Ethiopia. Roasted and ground barley is used as a major substrate during the fermentation stage [49]. This beverage also serves as a meal replacement for low income workers. Like other traditional Ethiopian fermented beverages (*Tella* and *Borde*), *Shamita* production does not require malt for the saccharification process [15].

To prepare *Shamita,* barley flour, salt, linseed flour, and a small amount of spice are mixed together with water to form a slurry liquid. As a starter culture, 1 to 2 L of previously produced slurry is added to the blend. The mixture is allowed to ferment overnight. Then, a small amount of bird’s eye chili (*C. annuum*) is added and the beverage is ready to serve for consumption [34].

The first full-length article on *Shamita* was published by Ashenafi and Mehari [34], which focused on the enumeration of microorganisms in samples collected from different vendors. The report found that lactic acid bacteria and yeasts are the dominant microorganisms in *Shamita*. Four years later, Bacha et al. [14] studied *Shamita* fermentation microbial dynamics and the microbial load of raw materials. Their study showed that barley is the major source of fermentative microorganisms. The count of these fermentative microbes reached 10^9^ CFU/mL after a 24 h fermentation period. Later Tadesse et al. [49] studied the antimicrobial effect of lactic acid bacteria isolated from *Shamita* on pathogenic microorganisms. The isolated lactic acid bacteria were found to inhibit the growth of the *Salmonella* species *S. flexneri*, and *S. aureus*. Similar inhibition was observed for lactic acid bacteria isolated from Nigeria’s oti-oka [8]. Additionally, the pH, conductivity, salinity and TDS values of *Shamita* were 3.8, 8391 µs/cm, 4.6% and 4520 mg/L, respectively [44].

### 2.4. Korefe

*Korefe* is a foamy fermented low alcoholic beverage popular in the northern and northwestern parts of Ethiopia. Similar to other Ethiopian fermented beverages, the fermentation system is natural and spontaneous. Barley, malted barley, “gesho”’ (*R. prinoides*), and water are the major ingredients used to prepare this indigenous beverage [50].

The process of making *Korefe* begins by mixing “gesho” (*R. prinoides*) and water to produce *“Tijit”* in a traditional container locally known as “*Gan”* (Figure 2). The blend is left for 72 h to extract flavor, aroma, bitterness and fermenting microorganisms [15]. While that is happening, non-malted barley powder is mixed with water to form a dough. The dough is then baked to make unleavened bread locally called *“Kitta”*. Then, “*Tijit”*, a small sized *“Kitta”* and an adequate amount of water are mixed together and left to ferment for about 48 h [39]. The semisolid mixture obtained at this stage is locally called *“Tenses”*. Subsequently, non-malted roasted barley powder, locally called *“Derekot”,* is added to the previously prepared *“Tenses”.* At this stage the blend is allowed to ferment for an additional 72 h. Finally, water is added to the mixture in a ratio of 1:3. After another 2 to 3 h of further fermentation the *Korefe* is ready to be served [15].

According to Getnet and Berhanu [15], the titratable acidity, ethanol, and crude fat content of *Korefe* are 32 g/L, 2.7% and 7.01%, respectively. In addition, the pH, conductivity, salinity and TDS values of *Korefe* are 3.7, 3199 µs/cm, 1.7% and 1610 mg/L, respectively [44]. After 72 h of fermentation, lactic acid bacteria and yeast counts were more than 10^9^ CFU/mL, whereas the *enterobacteriaceae* count was below the detectable limit due to the antagonistic effect of lactic acid bacteria [15].

### 2.5. Cheka

*Cheka* is a traditional low alcoholic fermented beverage commonly consumed in the southwestern parts of Ethiopia and particularly in Dirashe and the Konso district [36]. It is a cereal- and vegetable-based fermented low alcoholic beverage. Sorghum (*S. bicolor*), maize (*Z. mays*), finger millet (*E. coracana*), and vegetables such as leaf cabbage (*Brassica spp*.), moringa, (*Moringa stenopetala*), decne (*Leptadenia hastata*), and root of taro (*Colocasia esculenta*) are the main ingredients for *Cheka* preparation [17].

Worku et al. [35] reported a survey of raw materials and the production process of *Cheka*. According to their report, *Cheka* preparation starts by malting. The malt is prepared either from a single or a combination of the cereals listed above. Cabbage leaves and/or taro roots are cut into pieces and fermented anaerobically for about 4 to 6 d in a clean container. Then, a small amount of maize flour is added to the vegetable mixture and is fermented for an additional 2 to 3 d. The fermented vegetable mixture is then ground, filtered, and mixed with fresh maize flour. The fermentation continues for another 12 to 24 h. Then, water is added to the mixture and the mixture is allowed to ferment for one month. This fermented mixture is shaped into a dough ball, locally called “*Gafuma”,* and cooked at a temperature of 96 °C. After cooling, the cooked *“Gafuma”* is mixed with an adequate amount of previously prepared malt. The mixture is then allowed to ferment for an extra 12 h. This fermented mixture is locally called *“Sokatet”*. At this stage of the process a very thick porridge, locally called *“koldhumat”*, is prepared from maize flour. The prepared porridge is added to the vessel containing *“Sokatet”* with a sufficient amount of water. Finally, the mixture is left to ferment for another 4 to 12 h and served to consumers as *Cheka*.

Worku et al. [35] also published a paper that focused on the nutritional and alcohol content of *Cheka*. This report contained the physicochemical properties, ethanol, and methanol content of *Cheka* collected from *Cheka* producers. The average pH, ethanol, iron (Fe) and calcium (Ca) contents of *Cheka* samples are 3.76, 6%, 0.2 mg/g and 0.14 mg/g, respectively.

### 2.6. Keribo

*Keribo* is another alcoholic traditional beverage consumed by many Ethiopians, especially by those who prefer low alcoholic drinks. The production process is relatively less complicated [51].

Abawari [34] reported the raw materials and processing conditions of *Keribo*. According to the report, making *Keribo* begins by mixing roasted barley with hot water. Then, the mixture is boiled for about 20 min, after which the solid residue is removed by filtration. Subsequently, sugar and bakery yeast are added into the separated filtrate and left overnight to ferment. Finally, extra sugar is added to the mixture and the beverage is served to the consumer.

Abawari [16] published a second report that dealt with the microbial dynamics of *Keribo* fermentation. Based on the findings, average lactic acid bacteria, aerobic mesophilic bacteria, aerobic spore formers and yeasts counts were 2.70, 2.34, 4.96 and 2.01 log CFU/mL, respectively. However, the average *enterobacteriaceae*, *staphylococci*, and mold counts were below the detectable levels. Additionally, the shelf life of *Kerbio* is not more than two days at room temperature storage [40].

## 3. Fruit-Based Traditional Alcoholic Beverages

### 3.1. Tej

*Tej* is an Ethiopian wine that uses honey as a substrate and *“gesho”* (*Rhamnus prinoides*) as a source of bitterness. Previously, *Tej* was produced and consumed only for cultural festivities and for the royal families [52]. These days, *Tej* is a popular drink in rural, semi-urban, and urban areas of Ethiopia. It is produced and sold at the household level. The final product usually lacks consistency in quality due to differences in the manner of preparation and the ratio of ingredients used [21].

Ethiopia has the potential to produce 500,000 tons of bee honey annually. However, production has not surpassed 10% of that potential [53]. About 80% of the total honey produced in the country serves as raw material for producing *Tej* [54]. Traditionally, crude honey rather than refined honey is preferred for the production of *Tej* due to the distinct sensorial properties that local consumers prefer [18].

The *Tej* making process begins by cleaning and drying the traditional fermenting container. Then, honey and water are mixed in a ratio of 1:3 and allowed to ferment for 2 to 3 d. Afterwards, leaves and stems of *“gesho”* (*R. prinoides*) are boiled, cooled to room temperature and added to the previously fermented honey and water mixture. This mixture is allowed to ferment for 8 to 10 more days during the hot season or 20 d during the cold season [52]. After the intended period of fermentation, the product is ready to serve to the consumer in a special glass, locally known as *“Berele”*.

The microorganisms involved in the fermentation process originate from the raw materials, equipment and utensils. Because of this, *Tej* fermentation is lengthy, spontaneous, and uncontrolled. Thus, the final product have inconsistent physicochemical properties, microbiological profile, and sensory attributes [21].

Good quality *Tej* is yellow, sweet, fizzy, and cloudy due to the presence of active yeasts [43]. The flavor of *Tej* is highly dependent on the type of honey used and amount of “gesho” (*R. prinoides*) added. Additionally, the diversity and population of microorganisms also contribute to the distinctive flavor of *Tej* [55]. Like Mexican pulque [12], the Ethiopian *Tej’s* microorganism community is dominated by Lactic acid bacteria (LAB) and yeasts (Table 4). The shelf life and keeping quality of *Tej* is very short [40].

### 3.2. Ogol

*Ogol* is another traditional fermented honey wine beverage commonly consumed in the western part of Ethiopia. The preparation process starts by pulverizing the bark of the native tree *“Mange”* (*B. unijungata*). The pulverized bark, wild honey, and water are mixed in a container and the mixture is allowed to ferment for about two weeks. After completing the intended period of fermentation a small amount of water is added and the mixture is allowed to ferment anaerobically in a hot place for additional 12 to 36 h. Finally, it is filtered through a clean cloth and served to consumers as *Ogol* [19].

### 3.3. Booka

*Booka* is a low alcoholic traditional beverage that is popular in southern Oromia, Ethiopia (Table 4). The preparation process is relatively simple and easily adaptable. First the bladder of a cow is carefully removed from a dressed carcass and cleaned properly to remove residue urine. Honey and water are added to the prepared cow bladder in a ratio of 1:4. After 2 to 3 d of fermentation, a small amount of honey is added to the mixture and it is left to ferment anaerobically for an additional 2 d [43]. After the fermentation process is completed, the filtrate is ready to be served to consumers as *Booka*. Good quality *Booka* is yellowish in color, sweet in taste, and attractive in odor [20].

## 4. Nutritional Value, Function Properties and Safety Issues of Ethiopian Alcoholic Beverages

The nutritional values of Ethiopian traditional alcoholic beverages can be seen in two ways. In low alcoholic beverages, the nutritional values are higher than their respective raw materials [29]. The main justification forwarded by authors is the live microorganisms present in these beverages [14,34,47]. In high alcoholic beverages, the nutritional values are lower than that of low alcoholic traditional beverages [1,18]. As shown in Table 5, *Borde*, *Shamita* and *Cheka* have a good nutritional value compared to that of high alcoholic beverages like *Tella* and *Tej*. As the fermentation continues, from the fermentation dynamics point of view, only limited microorganisms withstand the adverse environmental effect of the growth medium. Thus, the microorganisms that do not cope with the new environment will be lysed and become a source of protein for cell maintenance for the surviving species. This analysis works even better in natural, spontaneous and uncontrolled fermentation systems. Hence, this competition in return decreases the nutritional value of the beverages while increasing secondary metabolites like ethanol [56,57,58].

The functional properties of the beverage are manifested in the content of total polyphenols (TP) and antioxidant activity (AA) [59]. These polyphenols and antioxidants have a health-promoting effect by scavenging free radicals and regulating metabolism [60]. Many Ethiopian alcoholic beverages have good TP and AA values (Table 5). The phenolic content of *Tella* is greater than that of *Korefe* and *Tej* [61]. Even though there are many factors responsible for this difference, raw materials, and especially the amount of gesho added to the mixture, take the lion’s share of the contribution [38].

The safety issues of Ethiopian traditional alcoholic beverages should be understood from the perspective of microorganism growth, higher alcohol and fluoride contents. Although the presence of a large amount of live fermentative microorganisms in low alcoholic traditional beverages contributes to their good nutritional value, there are major concerns related to food safety [17]. The microbiological safety issues were discussed in the previous section of this paper. This section focuses only on food safety issues related to higher alcohol and fluoride content. Higher alcohols contents of Isobutanol, 1-Butanol, 2-Butanol and 1-Propanol can be called collectively fusel oil or fuselol [62]. Fusel oil in a minute quantity contributes to the good flavor of the product. However, if it is consumed at a level above 1000 g/hL of pure alcohol, fusel oil is harmful for health [63]. The higher methanol content in traditional beverages also has a negative health impact [44]. Most of the time methanol is formed due to natural, spontaneous and uncontrolled fermentation [18]. As shown in Table 5, the methanol content of *Tella* and *Cheka* is very much lower than the maximum standard set by the European Union (EEC No 1576/89). Since Ethiopia is located in the region of the Great Rift Valley, fluoride ion concentration is another important food safety concern in traditional alcoholic beverages. A level of fluoride ions above 1.5 mg/L in the beverage creates dental and skeletal fluorosis [64]. Traditional beverages collected near Rift Valley localities showed a higher fluoride ion concentration (Table 5).

## 5. Conclusions and Future Perspectives

The most commonly produced and consumed Ethiopian traditional alcoholic beverages are *Tella*, *Borde*, *Shamita*, *Korefe*, *Cheka*, *Keribo*, *Tej*, *Ogol* and *Booka*. The ingredients, ratios, procedures and equipment used to prepare these beverages vary from place to place, but they all are produced through natural and spontaneous fermentation processes. Low alcoholic Ethiopian beverages have a higher nutritional value. Thus, they can be used as a meal replacement. These traditional alcoholic beverages also contain a significant amount of total polyphenols and antioxidants. The alcohol content and pH values of these beverages range from 1.53–21.7% and 2.9–4.9, respectively. As the fermentation continues, counts of lactic acid bacteria and yeasts species flourish while mesophilic aerobic bacteria and coliform counts decrease significantly. The source of microorganisms responsible for fermentation is mainly from the ingredients and utensils. These traditional alcoholic beverages show inconsistent quality within and between productions, and have a short shelf life. This is due to the high number of live cells present in freshly produced beverages.

Until now, research on Ethiopian traditional fermented beverages has mainly focused on the identification of raw materials and traditional processing methods. Moreover, microbial characterization and microbial dynamics have been reported for the last two decades. All of the reports have used culture-dependent phenotypic characterization. Hence, the current findings lack the completeness needed to lead these traditional beverages, which hold equal local market share with commercial products, into large-scale production. Thus, we find that future research has to shift its gear to a higher level by studying microbial metagenomics, starter culture development, rheological study, shelf life extension, process modification, kinetics, modeling and optimization.

## Figures and Tables

**Figure 1 foods-09-01781-f001:**
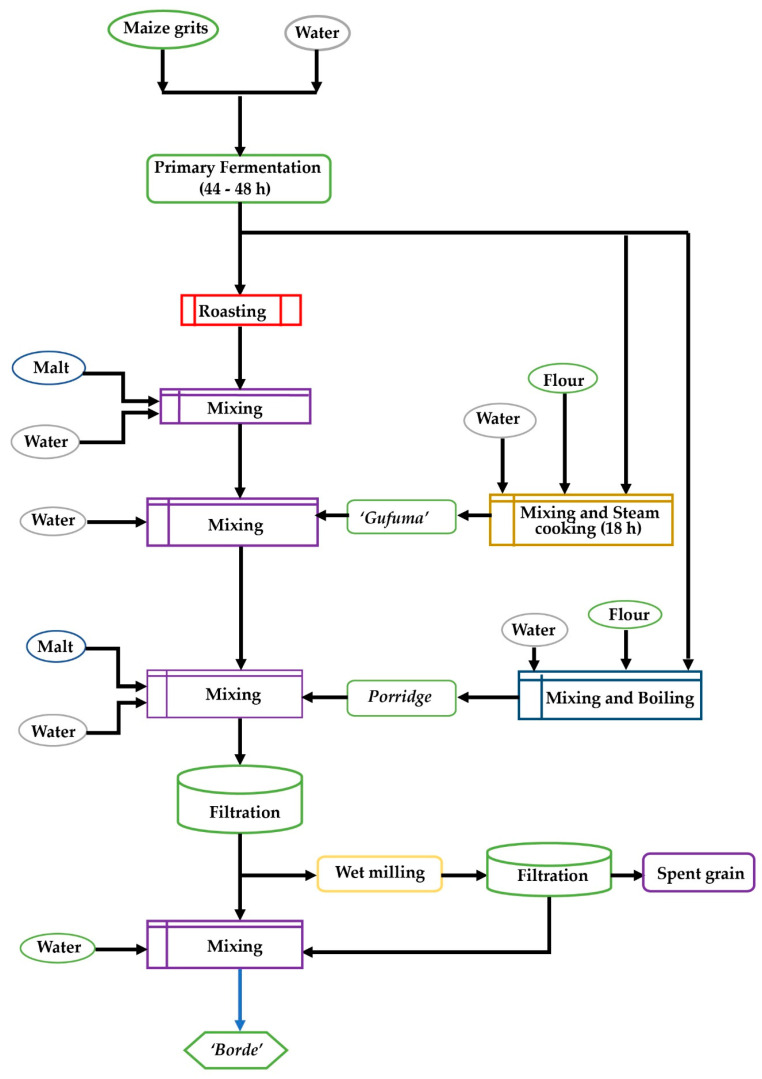
*Borde* processing flow chart [29].

**Figure 2 foods-09-01781-f002:**
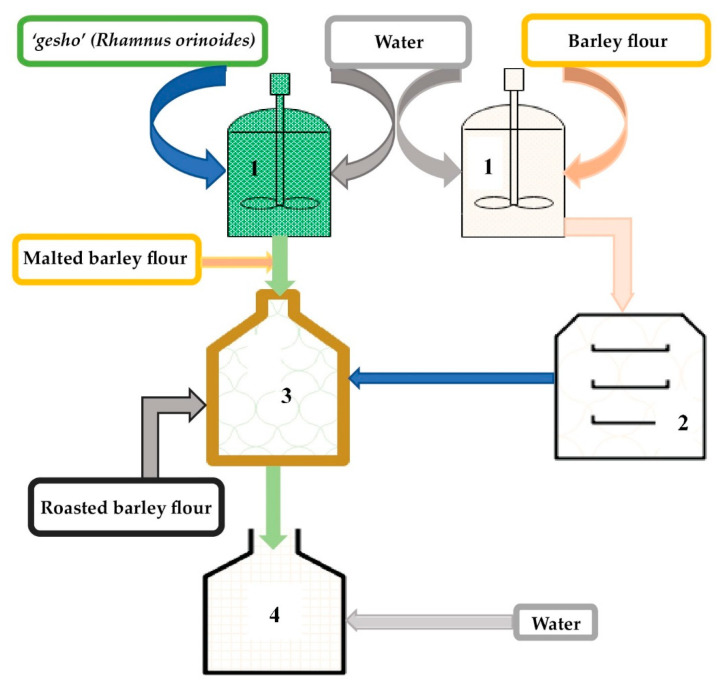
*Korefe* production process flow diagram: (**1**) mixer, (**2**) baking oven, (**3**) primary fermentation tank, (**4**) secondary fermentation tank.

**Table 1 foods-09-01781-t001:** Summary of cereal- and fruit-based Ethiopian traditional fermented alcoholic beverages.

Category of Beverages	Beverages	Raw Materials	Prominent Production and Consumption Regions	References
Beers	*Tella*	Barley (*Hordeum vulgare* L.), wheat (*Triticum aestivum* L.), maize (*Zea mays* L.), finger millet (*Eleusine coracana* L.), sorghum (*Sorghum bicolor* L.), *“teff”* (*Eragrostis tef* L.), “gesho” (*R. prinoides*)	Amhara, Oromia, Tigray, SNNP, Addis Ababa	[30,31]
	*Borde*	Maize (*Z. mays*), barley (*H. vulgare*), wheat (*T. aestivum*), finger millet (*E. coracana*), sorghum (*S. bicolor*)	SNNP	[13,32]
	*Shamita*	Roasted barley (*H. vulgare*) flour, salt, linseed (*Linum usitatissimum* L.) flour, chili pepper (*Capsicum annuum*)	SNNP, Addis Ababa	[13,33]
	*Korefe*	Malted and non-malted barley (*H. vulgare*), “*gesho*” (*R. prinoides*)	Amhara	[15]
	*Keribo*	Barley (*H. vulgare*), sugar, bakery yeast (*Saccharomyces cerevisiae*)	Oromia, Amhara, Addis Ababa	[16,34]
	*Cheka*	Sorghum (*S. bicolor*), maize (*Z. mays*), finger millet (*E. coracana*), vegetables, root of taro (*Colocasia esculenta* L.)	SNNP	[17,35,36]
	*Areke*	Barley (*H. vulgare*), “gesho” (*R. prinoides*), sorghum	Amhara, Oromia, Tigray, SNNP, Addis Ababa	[37,38,39]
Wine	*Tej*	Honey, “gesho” (*R. prinoides*)	Oromia, Amhara, Tigray, Addis Ababa	[18,39,40]
	*Ogol*	honey, barks of native tree (*Blighia unijungata* L.)	Gambella (Majangir)	[19]
	*Booka*	Honey, bladder of cow	Oromia (Gujii)	[20]

SNNP—Southern Nations, Nationalities, and Peoples Region.

**Table 2 foods-09-01781-t002:** Physicochemical properties, microbial load, and storage stability of *Tella.*

Area of Investigation	Shelf Stability, Microbial and Physicochemical Properties	Concluding Remarks	References
Storage stability, and microbial dynamics for vacuum filtered (VF), pasteurized (P) and control *Tella*	S. cerevisiae and Acetobacter xylinum (A. xylinum) are the dominating microorganisms; pH of control sample decreased, while VF and P pH samples increased during storage time; Turbidity of the control sample increased, while VF and P turbidity decreased or remained the same.	Pasteurization is an efficient method to extend the shelf life compared to vacuum filtration	[1]
Optimization of *Tella* production	3:1 (malt to “gesho” (*R. prinoides*)) showed lower pH after nine days of fermentation; Fermentation rate increased with increasing malt to “gesho” (*R. prinoides*) ratio.	The optimum fermentation process parameters: Temperature = 20–25 °C, average; pH = 4.78; Malt to “gesho” (*R. prinoides*) ratio = 1:3.	[46]
Isolation and characterization of *S. cerevisiae* from *Tella*	Six *S. cerevisiae* strains were isolated and characterized phenotypically; Isolates produce 10–15% mL/L of absolute ethanol; Isolates showed 84% of viability at higher sugar concentration; Isolates had an average 65% flocculation capacity.	Isolated strains have a good fermentative potential, especially for beer production	[30]
Physicochemical properties of fresh and matured *Tella*	pH 4.67–3.87; Alcohol content (%*v*/*v*) 3.04–3.75; Specific extract 1.0056–1.0037; Original extract 7.50–7.27; CO_2_ content (%) 0.24–0.034.	Alcohol content increases with increasing maturation time; pH, specific extract, original extract, and CO_2_ content decrease with increasing maturation time.	[31]

**Table 3 foods-09-01781-t003:** Processing methods and microbiological properties of *Borde.*

Area of Investigation	Microbial Load, Microbial Dynamics, and Processing Methods	Concluding Remarks	References
Isolation and characterization of lactic acid bacteria (LAB) involved in *Borde* fermentation	Heterofermentative *lactobacillus* (79.4%) is the predominate microorganism in *Bordeˆ;* Dominant species are: *Weissella confusa* (30.9%), *Lactobacillus viridescens*, (26.5%), *Lactobacillus brevis* (10.3%) and *Pediococcus pentosaceus* (7.4%).	Dominant microorganisms have a potential to be used as a starter culture	[13]
Antagonist effect of lactic acid bacteria over pathogenic microorganisms	■ At the beginning of fermentation Esherichia coli (E. coli O157:H7), Staphylococcus aureus (S. aureus), Shigella flexneri (S. flexneri) and Salmonella species counts are greater than 10^7^ CFU/mL; ■ After 16 h of fermentation the count of Salmonella species is less than log 2 CFU/mL, and all pathogenic microorganisms are below the detectable limit after 24 h of fermentation.	Secondary metabolites of LAB have a significant antimicrobial effect	[48]
Modified process technology for *Borde* production	Maize flour is substituted by maze grits; Remove wet milling from last stage of the process.	*Borde* making process can be simplified without compromising quality	[47]
Survey on local methods of processing and sensory analysis of *Borde*	■ Developed a traditional processing method with four-stage flow charts; ■ Maize, wheat, finger millet and sorghum used as raw materials; ■ Shelf life is no longer than 12 h at room temperature storage.	*Borde* has short shelf life; Production process is time-consuming.	[29]
Microbial dynamics of *Borde* fermentation	*Enterobacteriaceae* and coliform decreased from 10^4^ CFU/mL to below the detectable limit after 8 h of fermentation; Lactic acid bacteria increased from 10^6^ to 10^9^ CFU/mL within 24 h of fermentation time; Total fermentative yeast increased from 10^5^ to 10^7^ CFU/mL after 24 h fermentation time.	Yeast biota is dominated by *Saccharomyces* species; Keeping quality of *Borde* is very short.	[32]

**Table 4 foods-09-01781-t004:** Physicochemical and microbiological properties of *Tej*, *Ogol* and *Booka.*

Area of Investigation	Microbial and Physicochemical Properties	Concluding Remarks	References
Flora of yeast and lactic acid bacteria of *Tej*	*S.cerevisiae* (25%), *K. bulgaricus* (16%), *D. phaffi* (14%) and *K. veronae* (10%) are dominant yeast species; Lactic biota is composed of *Lactobacillus*, *Streptococcus*, *Leuconostoc* and *Pediococcus* species.	Yeasts and LAB are among the dominant microbes in *Tej* fermentation	[40]
Physicochemical properties of *Tej*	pH values of collected samples ranged between 3.07 and 4.90; Titratable acidity of samples ranged between 1 g/L and 1.03 g/L; Total alcohol content ranged between 2.7% and 21.7%; Average total dissolved solids (TDS) is 387%; Average electrical conductivity is 811 µs/cm; Average Salinity is 0.4 mg/L.	Natural and spontaneous fermentation is a major source of physicochemical variation in collected *Tej* samples	[18,44]
Isolating fermentative yeast from *Ogol*	*S. cerevisiae* is isolated from *Ogol* sample; Isolated species produce *16.5%* (*v*/*v*) ethanol; Titratable acidity and pH are 60 g/L and 3.8, respectively.	Isolated yeast from *Ogol* has the potential to be used for ethanol fermentation	[19]
Physicochemical properties of *Booka*	pH value ranges from 2.90–3.12; Moisture content—82.18%; Ash content—0.82%; Crude fat content—1.43%; Total nitrogen—7.01%; Total carbohydrate—8.56%; Mean alcoholic content—1.53%.	*Booka* can be used as a meal replacement	[20]

**Table 5 foods-09-01781-t005:** Nutritional value, functional properties and safety issues of Ethiopian alcoholic beverages.

Beverages	Nutritional Value	Functional Properties (Average Values)	Higher Alcohol and Fluoride Ion (Average Values)	References
*Tella*	Total protein—0.4% Carbohydrate—1.98%	TP (µg mL^−1)^—232.40 AA (µg mL^−1)^—296.00 Folate (mgcg^−1^)—0.093	Fusel oil (ppm)—51 Methanol (ppm)—41.5 Fluoride ion (mg/L)—4.26	[1,46,61,65]
*Borde*	Total protein—9.55%, Crude fat—6.88%, Total ash—3.66%	TP (µg mL^−1)^—9.50 AA (µg mL^−1)^—198.5	Fluoride ion (mg/L)—4.95	[29,33,39,65]
*Shamita*	Total protein—10.37% Crude fat—6.85% Total ash—3.46%	–	Fluoride ion (mg/L)—5.21	[33,34,65]
*Korefe*	–	TP (µg mL^−1)^—167.60 AA (µg mL^−1)^—278.13	Fluoride ion (mg/L)—1.39	[63,65]
*Cheka*	Total protein—3.83% Crude fat—1.49% Carbohydrate—16.59% Total ash—0.79%	–	Methanol(ppm)—271.55	[17]
*Keribo*	–	TP (µg mL^−1)^—12.65 AA (µg mL^−1)^—64.66	–	[38]
*Tej*	Total protein—0.35% Crude fat—0.35% Carbohydrate—3.58% Total ash—0.04%	TP (µg mL^−1)^—197.00 AA (µg mL^−1)^—240.37	Fusel oil (ppm)—205.08 Fluoride ion (mg/L)—6.68	[18,21,38,61,65]
*Bokaa*	Moisture content—82.18% Ash content—0.82% Crude fat content—1.43% Total Nitrogen—7.01%	–	–	[20]

TP in gallic acid equivalent (GAE); AA in ascorbic acid equivalent (AAE); —values not available in the literatures.

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
