# Peer review of "Cereal- and Fruit-Based Ethiopian Traditional Fermented Alcoholic Beverages"

_foods, 2020, doi:10.3390/foods9121781_

Round 1

Reviewer 1 Report

The review deals with the microbial and technological description of some Ethiopian production processes of traditional fermented alcoholic beverages. Topic is a novelty and the reported data could be useful for comparing with similar processes from other countries. Processes are described in a very detailed way but in incorrect form. Moreover, a spell and grammar check must be done by the Authors to improve the English and the comprehension of the manuscript.

Point 1: Change all the verb tense in the description of all the processes. They are not recipes.

i.e.

  • L74: not “cover the moisten grain” but “the moisten grain is covered”
  • L87: not “add water to Difdif” but “water is added to Difdif

and so on.

Point 2: Check the English very careful

i.e.

  • L15: “beverages are drinks produced locally” instead of “beverages are drink, which produced locally”
  • L22: uniform the verb tense
  • L22: “yeasts” instead of “yeast”
  • L47: add “be” after “can alternatively”
  • LL49-50: reformulate the sentence
  • L58: “outputs” instead of “out puts”
  • L101: “sour” instead of “sore”
  • L110: “liters” instead of “litters”
  • LL192-193: reformulate this sentence
  • L226: “tonnes” instead of “tones”
  • LL242-243: reformulate
  • L243: “instinct”??

and so on.

Point 3: Add reference to tables and figures in the text.

Other minor revision:

  • The use of the word “microbe” (or “microbes”) is not a mistake, but talking about food fermentation is better to use the word “microorganism” (or “microorganisms”)
  • Title and L17: “cereal- and fruit-based” instead of “cereal and fruit based”
  • LL38-39, 41-42: the name of the products should be in Also in some other section.
  • Check the double space, i.e. LL38, 70...
  • Check “barely” and “barley”
  • Check “prenoids” and “prinoides
  • The name of the microorganisms must be in italic. i.e. LL92, 94, 101...
  • L97: specify the unit of measure of antioxidant activity. Only µg/mL-1 is not enough
  • Check and uniform “ml” or “mL”; “CO2”; “25°C” or “25 °C”...
  • Table 2: “hop” is the same of “gesho” in this case?
  • Table 3: first time you use “LAB”, the same for “ coli”, “S. aureus”...
  • L184: are you sure that “decne” is the name of Leptadenia hastate? Or maybe the classification
  • Check all the units of measure, also in the tables
  • LL255-256: “remove the bladder of cow”, from where?

Reviewer 2 Report

*The review topic is interesting and may bring important information on the Ethiopian beverages, but the information should be better structured and completed with much more information on biochemical and microbiological characteristics of the traditional products.
*Considering that it is a review, the References may be completed with information from much more authors like:
Tafere, 2015. A review on Traditional Fermented Beverages of Ethiopian. Journal of Natural Sciences Research
Bikila Wedajo Lemi, "Microbiology of Ethiopian Traditionally Fermented Beverages and Condiments", International Journal of Microbiology, vol. 2020, Article ID 1478536, 8 pages, 2020. https://doi.org/10.1155/2020/1478536
Shewakena, S.; Chandravanshi, B. S.; Debebe, A. 2017, Levels of total polyphenol, flavonoid, tannin and antioxidant activity of selected Ethiopian fermented traditional beverages. International Food Research Journal. Oct2017, Vol. 24 Issue 5, p2033-2040. 8p.
*The review is recommended to have in the beginning a Table of content.
* All tables and figures must be cited in the text
* The manuscript needs to have a Discussion section related especially to food safety, nutritional values and functional potential.
* Other observations are provided directly on the manuscript.

Round 2

Reviewer 1 Report

Authors improved the manuscript following almost all the Reviewers suggestion. The English could be still improved. For this reason I recommend minor revision before the acceptance.

Some revisions:

  • Ok, the Authors substituted “cereal and fruit based” with “cereal- and fruit-based” in the title and in other part of manuscript, but they should check all other sections, especially the new one (i.e. LL 34, 41, 87, Table 1 title, and so on). Moreover add a space between “cereal-“ and “and”
  • Again, the name of all products should be in i.e. LL 58-59, 61-62, 122-123, 152, 227
  • Check again the double space, i.e. LL 58, 96, 124, 220, 221, 270, 271, 394...
  • L67: Check the reference citation [22]. It’s not Henry
  • LL68 and 70: “as” instead of “is”
  • L72: add “are” after “cane and palm”
  • L80: “their” instead of “its”
  • L102: “barley” instead of “barely”
  • Table 2: Check name in italic (xylinum), °C, mL, double space, capital letters, add “pH” between “VF & P” and “samples”, add “turbidity” between “VF & P” and “decreased”
  • Table 3: Check name in italic (lactobacillus, enterobacteriaceae), 12h, maze, microbes, use “biota” instead of “flora”
  • L200: add “acid” between “lactic” and “bacteria”
  • L202: “species” (or spp.) should be not in italic
  • L220 and Table 4: specify the unit of measure of titratable acidity. Why “%”?
  • L222: remove “,” between “bacteria” and “and yeast counts”
  • L222, L265: “enterobacteriaceae” in italic
  • LL252-253: reformulate this sentence
  • L256: remove “.” between “drink” and “s”
  • L293: “yeasts” instead of “Yeasts”
  • L300: add “.” after “consumers”
  • L301: “hotter”? hotter rather than what?
  • Table 4: use “biota” instead of “flora”, check some values in italic, “composed by Lactobacillus” instead of “composed of Lactobacillus
  • LL338-242: reformulate these sentences
  • LL349-351: reformulate this sentence
  • L360: “hL” instead of “hl”
  • L391: “mainly” instead of “manly”
  • LL233-234: Sorry, but I’m not agree with the Aurhors. “Decne” is the name abbreviation of the Author (Joseph Decaisne) that classified Leptadenia hastata (not hastate). The Authors can see that also in their web link.

Author Response

Dear Reviewer 1, we would like to say thank you very much, for your constructive comments. We really appreciate your deep review. We agree with all of your comments and we have revised the manuscript accordingly. Please find below our point by point responses (in red). Beside, the additional English style and spells were corrected by a correction company.

Point 1: Ok, the Authors substituted “cereal and fruit based” with “cereal- and fruit-based” in the title and in other part of manuscript, but they should check all other sections, especially the new one (i.e. LL 34, 41, 87, Table 1 title, and so on). Moreover add a space between “cereal-“ and “and”

Response 1: We acknowledged your comment and it is now corrected in the manuscript (Page 1: L 2,18,34), (Page 2: L 41), (Page 3: L 88), (Page 6: L 140), (Page 11: L 266), (Table 1)

Point 2: Again, the name of all products should be in i.e. LL 58-59, 61-62, 122-123, 152, 227 Check the English very careful

Response 2: we highly appreciate your comment. Now, it is corrected in the manuscript as Rwanda’s Ikigage [7] (Page 4: L 123), Korean’s makgeolli [11] (Page 4: L 124), Uganda’s kwete [9] (Page 6: L:151), Kenya’s Busaa [10] (Page 6: L 165), Nigeria’s oti-oka [8] (Page 9: L 201), Mexican’s pulque [12], (Page 12: L 291) And for Ethiopian beverages … microorganisms are the ingredients to produce Tella [1].  (Page 3: L 93), Borde making process starts … (Page 6: L 147), Bacha et al. [14] studied Shamita fermentation microbial dynamics …..(Page 9: L 196) After another 2 to 3 h of further fermentation the Korefe is ready to be served [15] (Page 9: L 218)  Abawari [16] published a second report that dealt with on the microbial dynamics of Keribo fermentation. (Page 11: L 261) … are the main ingredients for Cheka preparation [17](Page 10: L 234) … preferred for the production of Tej due to the distinct sensorial properties that local consumers prefer [18] (Page 11: L 276,277) Finally, it is filtered through a clean cloth and served to consumers as Ogol [19] (Page 12: L300) Good quality Booka is yellowish in color, sweet in taste, and attractive in odor [20] (Page 12: L 308)                 

Point 3: Check again the double space, i.e. LL 58, 96, 124, 220, 221, 270, 271, 394...

Response 3: we acknowledge your comment and it is now corrected 

Point 4: L67: Check the reference citation [22]. It’s not Henry

Response 4: we highly appreciate your critical review and we corrected as Herman [22] (Page 2: L 67)

Point 5: LL68 and 70: “as” instead of “is”

Response 5: we very much acknowledge your comment and we corrected as Scholars define wine and beer based on various perspectives. For instance, Herman [22] define wines as alcoholic beverage made from sound ripe grapes. Whereas, Pederson [23] defines alcoholic beverages based on the kind of substrates: beers are produced from cereals while wines are produced from fruits. Beside, Steinkraus [24] define wine as alcoholic beverage that uses sugar as the principal source of fermentable carbohydrate (Page 2, L 67)

Point 6: L72: add “are” after “cane and palm”

Response 6: we again thank for your comment and now, it is corrected as “cane and palm are classified under the category of wine” (Page 2: L 72)

Point 7: L80: “their” instead of “its”

Response 7: Thank you very much, we corrected in the manuscript (Page 2: L 80)

Point 8: L102: “barley” instead of “barely”

Response 8: Thank you, we again corrected in the manuscript (Page 4: L 103)

Point 9: Table 2: Check name in italic (xylinum), °C, mL, double space, capital letters, add “pH” between “VF & P” and “samples”, add “turbidity” between “VF & P” and “decreased”

Response 9: thank you. Table 2 is corrected as per the comments

Point 10: Table 3: Check name in italic (lactobacillus, enterobacteriaceae), 12h, maze, microbes, use “biota” instead of “flora”

Response 10: Again, thank you, we corrected Table 3 as per the comments

Point 11: L200: add “acid” between “lactic” and “bacteria”

Response 11: Again, thank you, we corrected it (Page 9, L 199)

Point 12: L202: “species” (or spp.) should be not in italic

Response 12: Thank you very much, we corrected it (Page 9, L 200)

Point 13: L220 and Table 4: specify the unit of measure of titratable acidity. Why “%”?

Response 13: we very much acknowledged your knowledge-based comment and we corrected to g/L (Page 9, L200) & Table 4

Point 14: L222: remove “,” between “bacteria” and “and yeast counts”

Response 14: Thank you very much, we corrected it (Page 9, L 222)

Point 15: L222, L265: “enterobacteriaceae” in italic

Response 15: Again, thank you, we corrected it (Page 9, L 222, 264)

Point 16: LL252-253: reformulate this sentence

Response 16: we appreciate your comment and corrected in the manuscript as the average pH, ethanol, Iron (Fe) and calcium (Ca) content of Cheka samples are 3.76, 6%, 0.2 mg/g, 0.14 mg/g, respectively (Page 11: L 251-252)

Point 17: L256: remove “.” between “drink” and “s”

Response 17: Thank you very much, we corrected it (Page 11: L 255)

Point 18: L293: “yeasts” instead of “Yeasts”

Response 18: Thank you very much, we corrected it (Page 11: L 292)

Point 19: L300: add “.” after “consumers”

Response 19: Thank you very much, we corrected it (Page 12: L 298)

Point 20: L301: “hotter”? hotter rather than what?

Response 20: Thank you very much, we corrected in the manuscript as “… the mixture is allowed to ferment anaerobically in a hot place for additional 12 to 36 hours.” (Page 12: L 299)

Point 21: Table 4: use “biota” instead of “flora”, check some values in italic, “composed by Lactobacillus” instead of “composed of Lactobacillus

Response 210: Thank you very much, we accepted your valuable comment and corrected accordingly (Page 13: Table 4)

Point 22: LL338-242: reformulate these sentences

Response 22: Again, thank you very much, we corrected as As the fermentation continues, from the fermentation dynamics point of view, only limited microorganisms withstand the adverse environmental effect of the growth medium. (Page 14: L 335-337)

Point 23: LL349-351: reformulate this sentence

Response 23: Thank you very much, we corrected as even though there are many factors for this difference, raw materials especially the amount of gesho added to the mixture, takes the lion’s share of the contribution [39]   (Page 14: L 346-347)

Point 24: L360: “hL” instead of “hl”

Response 24: Thank you very much, we corrected accordingly (Page 14: L 356)

Point 25: L391: “mainly” instead of “manly”

Response 25: Again, thank you very much, we corrected (Page 16: L 388)

Point 26: LL233-234: Sorry, but I’m not agree with the Aurhors. “Decne” is the name abbreviation of the Author (Joseph Decaisne) that classified Leptadenia hastata (not hastate). The Authors can see that also in their web link.

Response 26: At last, we would like to thank for your critical observation. Now it is corrected as decne (Leptadenia hastata)

Reviewer 2 Report

No further comments and suggestions. The authors have answered satisfactorily to all the questions/requests.

Author Response

Thanks for your efforts, the English style and spells are corrected with our best efforts.